

# Impact of P fertilizer and arbuscular mycorrhizal fungi on forage legume growth, chlorophyll content and productivity

Sanele Mpongwana[1], Alen Manyevere[2], Conference Thando Mpendulo[1], Johnfisher Mupangwa[3], Wandile Mashece[4] and Mthunzi Mndela[4]

[1] Livestock and Pasture, University of Fort Hare, Alice, South Africa
[2] University of Fort Hare, Agronomy, Alice, South Africa
[3] Agribusiness & Economics, University of Namibia, Neudamm, Windhoek, Namibia
[4] Livestock and pasture, University of Fort Hare, Alice, Eastern Cape, South Africa

Corresponding author
Wandile Mashece,
wandilemashece@gmail.com

## ABSTRACT

Soil phosphorous (P) is the most limiting plant nutrient globally, reducing forage plant productivity. Although inorganic P fertilizers are used, about 75–90% of P becomes unavailable for plant uptake, hence, the strategies to enhance P uptake acquisition, such as the use of arbuscular mycorrhizal fungi (AMF) inoculation, are crucial. A greenhouse pot experiment was conducted under controlled environmental conditions at the University of Fort Hare, where three legume species (*Vigna unguiculata*, *Lablab purpereus* and *Mucuna pruriens*) were grown for 90 days under five P fertilizer levels (0; 0.68; 1.36; 2.04 and 2.72 g P/pot) with or without AMF-inoculation, resulting in 30 treatment factorial arrangement, each replicated 4 times. Agronomic responses to P fertilization and AMF-inoculation were assessed. Plant height, stem diameter, chlorophyll content, and leaf and stem yield were significantly influenced ($p < 0.001$) by the interaction of phosphorus (P) fertilizer levels, arbuscular mycorrhizal fungi (AMF) inoculation, and legume species. Inoculated plants showed remarkable growth, reaching heights of 94.2 to 159.0 cm compared to 61.1 to 117.0 cm in uninoculated plants. Additionally, inoculated plants had stem diameters twice as large as those of uninoculated plants when grown with 1.36 g P/pot, outperforming other P fertilizer levels by day 90 across all legume species. Likewise, chlorophyll content of inoculated plants (78.1–90.7 soil plant analysis development (SPAD)) was significantly higher than uninoculated plants (56.9–69.1 SPAD) at 1.63 P g/pot compared to 0, 0.68, 2.04 and 2.72 g P/pot. Moreover, inoculated plants attained relatively higher leaf (123.3–144.0 g/pot) and stem yield (75.2–121.8 g/pot) than uninoculated plants at 1.36 g P/pot compared to 0, 0.68, 2.04 and 2.72 g P/pot. Overall, AMF-inoculation improved growth and productivity of forage legumes, but its effects depended on the P fertilizer level, with 1.36 g P/pot being the potential optimum fertilizer rate for soil nutrition of legume pastures.

## INTRODUCTION

Low forage production is one of the constraints limiting livestock production in developing countries. This may be attributed to factors such as rangeland degradation, and climate change (*Mpongwana et al., 2023a*; *Mpongwana et al., 2023b*). Thus, the reliance on rangelands alone for livestock feeding is inadequate, given the ever-growing human population that demands more animal products. Legume pastures have the potential to complement rangelands as a source of highly nutritious and digestible forage (*Aucamp, 2008*). However, poor soil fertility, largely phosphorus (P) deficiency remains a central constraint to sustainable legume pasture establishment and productivity (*Mitran et al., 2018*; *Mpongwana et al., 2023a*; *Mpongwana et al., 2023b*). Soil P deficiency in legumes limits nodulation, and *Rhizobium* establishment, thereby reducing legume growth and productivity (*Mitran et al., 2018*). Additionally, soil P deficiencies have led to the reliance on inorganic P fertilizers (*Bastida et al., 2023*), with P accessed as orthophosphate anions ($H_2PO_4^-$ and $HPO_4^{2-}$), P forms restricted to soil pH of 6.0 to 6.5 (*Ibrahim et al., 2022*). Furthermore, only a small proportion (10–20%) of P applied as an inorganic fertilizer is available for uptake by plants (*Helfenstein et al., 2018*). P immobilization and insolubility are the main factors limiting P uptake, as P tends to be adsorbed by aluminium (Al) and iron (Fe) together with clay minerals, making it unavailable for plant uptake (*Bastida et al., 2023*; *Mpongwana et al., 2023b*).

This has stimulated more research interest in finding strategies to maximize P availability, uptake, and efficient use by forage plants (*Bastida et al., 2023*). Of these strategies, plant inoculation with arbuscular mycorrhizal fungi (AMF) holds great promise through its mutualistic relationship with plants *via* carbon-for-nutrient trade (*Ibrahim et al., 2022*). The AMF acquires carbon (C) from plants in exchange for P and to some degree nitrogen (N) (*Antunes et al., 2012*; *Nouri et al., 2014*). The enzyme phosphatase produced by AMF solubilizes immobile P, thereby increasing P availability and uptake by plants (*Begum et al., 2019*), and the extraradical mycelia formed by the fungi on plant roots grow beyond plant rooting depth to acquire soil nutrients (*Nouri et al., 2014*; *Ibrahim et al., 2022*).

AMF plays a vital role in tripartite mutualistic interactions with *Rhizobia* and the host plant, thereby enhancing the efficiency of $N_2$ fixation, water uptake and disease resistance and reducing the effects of environmental stresses *e.g.*, drought and heavy metal stress (*Hack et al., 2019*; *Murrell et al., 2020*). These together with increased P uptake promote photosynthesis, enhance plant health, thereby increasing leaf area index (LAI), and overall forage productivity of legumes (*Püschel et al., 2017*). The AMF-mediated benefits are more important given that P and N are the main limiting factors for plant growth and productivity (*Liu et al., 2021*; *Mpongwana et al., 2024*). Thus, the inclusion of AMF-inoculation in soil nutrient management and planting programs may reduce the costs of fertilizer application for financially disadvantaged farmers. While the initial cost of obtaining AMF-inoculants may be higher, it can result in long-term savings by reducing the need for repeated fertilizer applications. Moreover, fertilizers are often more expensive and need continuous replenishment, whereas AMF can establish in the soil and provide ongoing benefits at a lower cumulative cost.

A collaborative initiative on legume pasture establishment was launched in 2006 by the Eastern Cape government of South Africa and the Australian government to improve livestock production in communal areas (*Davis, Ainslie & Finca, 2008*). To the best of our knowledge, this was the first empirical attempt to establish legume pastures in communal arable lands of South Africa. Hence, little has been done to ascertain factors that limit legume pasture productivity including low availability of soil P. Previous studies demonstrate that the interactions of AMF and *Rhizobia* to influence legume productivity depend on the amount of P in the soil, with low and super high P reducing the benefits of AMF (*Püschel et al., 2017*). While there is plenty of evidence indicating that AMF enhances P availability and uptake to increase productivity (*Unger et al., 2021*), a knowledge gap exists regarding the rate of P fertilizer at which AMF maximizes legume growth and productivity. This information is crucial to designing an appropriate soil nutrition management program for sustainable legume pasture establishment and production. This study, therefore, answers the following questions: (1) does the influence of P fertilizer application on legume growth, chlorophyll content and productivity depend on the AMF-inoculation? (2) What is the optimal rate of P fertilizer at which AMF maximizes forage legume growth, chlorophyll content and productivity? (3) Do the growth and productivity responses of forage legumes to AMF and P fertilizer vary with legume species?

## MATERIALS AND METHODS

### Study site and experimental design

A greenhouse pot experiment was conducted at the University of Fort Hare (UFH), Alice Campus, Eastern Cape (32°46′S 26°50′E). The experiment commenced on 01-January 2017 and finished on 30-April 2017. Potting soil was collected from the agronomy section of the UFH Crop Research Farm, making sure to collect at a depth of 15 cm, with a bulk density of 1,550 g/cm³. An initial soil analysis was done at the beginning of the experiment, for the analysis of soil (soil pH, soil organic carbon, soil organic matter, soil N, soil available P, potassium (K), calcium (Ca), magnesium (Mg), Fe (Na), zinc (Zn), manganese (Mn), and copper (Cu) Table 1. The soil properties and nutrient analysis were determined according to the methods mentioned in the work of *Mpongwana et al. (2024)*.The experiment was designed as split-split pot design (SSPD) with a 3 × 2 × 5 factorial arrangement. The SSPD comprised of three legume species (*L. purpureus, M. pruriens* and *V. ingucualata*), two mycorrhizal inoculation levels, and five phosphorus rates were 0; 0.68; 1.36 and 2.04 g P/pot. Legume species were the main factor, arbuscular mycorrhizal fungi was subfactor, and phosphorus fertilizer rates were sub-subfactor. Phosphorus fertilizer rates were 0; 0.68; 1.36; 204 and 2.72 g P/pot which are the equivalents of 0; 20; 40; 60 and 80 kg P/ha. Conversions were done using the amount (two million kg soil) of soil in the top 15 cm of 1 ha area with soil bulk density 1,550 g/cm³. Each legume was planted in 40 pots, with 20 pots planted with inoculated seeds and the other 20 pots planted with uninoculated seeds. For both the pots of inoculated and uninoculated seeds, each of the P fertilizer rates was applied in four replicate pots. This resulted in 30 treatment combinations (3 species × 2 arbuscular mycorrhizal fungi levels × 5 fertilizer levels), giving a total of 120 pots.

**Table 1** The nutrient composition of the soils used for the experiment.

| Soil nutrient | Amount |
|---|---|
| pH (H20) | 6.9 |
| Organic carbon (%) | 0.65 |
| Organic matter (%) | 0.92 |
| Total nitrogen (%) | 0.066 |
| P (mg/kg) | 2.34 |
| K (mg/kg) | 44.7 |
| Ca (mg/kg) | 546.5 |
| Mg (mg/kg) | 183.5 |
| Fe (mg/kg) | 17.7 |
| Na (mg/kg) | 1.87 |
| Zn (mg/kg) | 34 |
| Mn (mg/kg) | 45 |
| Cu (mg/kg) | 40 |

The potting soil was sterilized, to kill potential arbuscular mycorrhizal fungi (AMF) that might be present in the soil, through oven drying at 82–92 °C for 30 min (*Ortas, 2012*). Before planting, seeds were inoculated with *Rhizobium* inoculum (*Bradyrhizobium* strain). A commercial AMF product (Mycoroot™ Supreme) was purchased from Rhodes University Microbiological Lab. A mixture of one ml of Mycoroot™ Supreme per seed was applied, followed by the application of single superphosphate fertilizer ($P_2O_5$). The rates of single superphosphate applied were 20 kg P/ha (0.68 g P/pot), 40 kg P/ha (1.36 g P/pot), 60 kg P/ha (2.04 g P/pot) and 80 kg P/ha (2.72 g P/pot) in a 15 kg soil per pot. The AMF comprised of various isolates including *Claroideoglomus etunicatum, Funneliformis mosseae, Gigaspora gigantean, Paraglomus occulum* and *Rhizophagus clarus* (*Mpongwana et al., 2023a*; *Mpongwana et al., 2023b*). The diameter of the pot was 30 cm and the soil depth in the pot was 30 cm (15 kg soil mass). The forages were not supplied with other fertilizer, except the single superphosphate fertilizer and a standard *Rhizobium* inoculation for all the forage legumes as a standard control.

Three seeds per pot were planted at 4–6 cm depth of soil and thinned to two after seedling emergency. All the pots were randomly placed in a greenhouse with temperatures of 27 °C with natural light. The cooling of the greenhouse was achieved by regulating the air condition. Irrigation was done once a day in the morning to maintain moisture at 50% field capacity to avoid leaching. This was achieved by measuring soil moisture content using calibrated soil moisture probes Delta T device (SM150T Soil Moisture Sensor, United Kingdom). The pots were kept weed-free through hand removal of any emerging weed.

## Data collection

The legume plant height (cm) was measured from the base to the tip of a primary shoot in all plants at 30, 60 and 90 days after sowing using a measuring tape. Stem diameter (mm) was measured in each plant at 10 cm above the soil surface using the vernier calliper (Mitutoyo, 150 mm vernier caliper 0.02 mm, Metric, Zhejiang, China) at 30, 60 and 90 days after sowing. The leaf chlorophyll content was measured using a soil plant analysis development

(SPAD) meter (SPAD-502 Plus, Minolta Camera Cooperative, Japan) (*Rodriguez & Miller, 2000*) at 15-day intervals starting from day 30 of sowing by randomly selecting three leaves on each plant per pot.

The forage legumes were cut at a stubble height of 10 cm above the ground, 90 days after sowing. The forage samples were bagged and transported to the lab where they were separated into leaf and stem, after which the fresh weight of each forage component was determined. Thereafter, forage samples were oven-dried at 65 °C for until constant weight was obtained. The leaf and stem weights were used to calculate the leaf-to-stem ratio.

### Statistical analysis

Firstly, data normality and homoscedasticity were assessed using Kolmogorov-Smirnof and Levenne's tests, respectively and all the data met these assumptions. Repeated measures analysis of variance (RMANOVA) using mixed effects models was conducted using SAS version 9.1.3 (SAS 2003), with time since sowing entered as within-subject factor, whereas legume species ($n = 2$), inoculation ($n = 2$) and P fertilizer levels ($n = 5$) were added as between-subject factors. When interactions were significant at $\alpha = 5\%$, the means were separated using Tukey's test. Pearson correlations were conducted to assess bivariate relationships using Holm adjustment method.

## RESULTS

### Shoot height and stem diameter

The interactions between P fertilizer, AMF-inoculation and legume species on plant height and stem diameter overtime since sowing are presented in Table 2. The interaction between legume species, P fertilizer levels and AMF had a significant ($p < 0.05$) effect on the plant height. For all legume species, inoculated plants were significantly ($p < 0.05$) taller (30.77–34.15 cm), more so under 1.36 g P/pot the value was equivalent to 40 P kg/ha fertilizer level compared to uninoculated plants (17.75–23.07 cm) 30 days post-sowing. However, when inoculated plants were compared alone, *V. unguiculata* plants were relatively taller under1.36 g P/pot the value was equivalent to 40 kg P/ha (30.77–86.72 cm) and 2.04 g P/pot (equivalent to 60 kg P/ha) (28.17–74.15 cm) after 30 and 60 days of sowing, after which the differences disappeared between 0, 0.68; 1.36, 2.04 and 2.72 on day 90 post-sowing. For *L. purpereus* and *M. pruriens*, inoculated plants had similar height at 1.36 and 2.04 g P/pot, but the plants under these P fertilizer levels were consistently taller than at 0, 0.68 and 2.72 g P/pot throughout the study period.

There were significant interactions ($p > 0.05$) of legume species, P fertilizer and AMF on stem diameter (SD), with stem diameter increasing with P fertilization, peaking at 1.36 g P/pot, above which it declined. There were obvious differences between inoculated and uninoculated plants for all legume species. The former exhibited a twofold larger stem diameter than the latter, more significantly ($p < 0.05$) at 0, 1.36 and 2.04 g P/pot after day 60 and 90 post-sowing.

### Chlorophyll content

The interactions between P fertilizer, AMF inoculation and legume species on chlorophyll content overtime since sowing are presented in Table 3. The three-way interactions of

**Table 2 Interactions between legume species, AM fungi inoculation and P fertilizer level on plant height and stem diameter over time since sowing.**

| | | | Time elapsed since sowing (days) | | | | | |
| | | | Plant height (cm) | | | Stem diameter (mm) | | |
| Independent variables | | | | | | | | |
| Species | AMF | Fertilizer (kg P/ha) | 30 | 60 | 90 | 30 | 60 | 90 |
|---|---|---|---|---|---|---|---|---|
| *V. anguiculata* | Uninoculated | 0 | 16.00$^d$ | 48.77$^e$ | 61.12$^f$ | 0.47$^{de}$ | 0.85$^b$ | 1.32$^d$ |
| | | 20 | 17.85$^d$ | 48.70$^e$ | 67.62$^{ef}$ | 0.60$^{de}$ | 0.95$^b$ | 1.80$^d$ |
| | | 40 | 23.07$^{bc}$ | 49.52$^e$ | 61.12$^f$ | 0.52$^{de}$ | 0.95$^b$ | 2.25$^{cd}$ |
| | | 60 | 26.40$^b$ | 44.67$^e$ | 62.37$^f$ | 0.57$^{de}$ | 0.97$^b$ | 1.82$^d$ |
| | | 80 | 15.75$^d$ | 41.32$^e$ | 66.47$^f$ | 0.55$^{de}$ | 0.97$^b$ | 1.60$^d$ |
| | Inoculated | 0 | 25.65$^b$ | 63.77$^d$ | 73.75$^{ef}$ | 1.22$^b$ | 1.67$^b$ | 2.42$^{cd}$ |
| | | 20 | 26.57$^b$ | 69.60$^c$ | 76.25$^{ef}$ | 1.27$^b$ | 1.67$^b$ | 2.62$^{cd}$ |
| | | 40 | 30.77$^a$ | 86.72$^b$ | 94.20$^e$ | 1.62$^a$ | 3.72$^a$ | 6.02$^a$ |
| | | 60 | 28.17$^{ab}$ | 74.15$^{bc}$ | 85.82$^e$ | 1.47$^{ab}$ | 2.55$^a$ | 4.12$^{bc}$ |
| | | 80 | 29.72$^a$ | 45.27$^e$ | 61.12$^f$ | 0.70$^d$ | 1.02$^b$ | 2.40$^{cd}$ |
| *L. purpereus* | Uninoculated | 0 | 9.85$^e$ | 22.25$^g$ | 88.77$^{de}$ | 0.77$^{cd}$ | 1.20$^b$ | 1.67$^d$ |
| | | 20 | 12.05$^{de}$ | 26.45$^{fg}$ | 99.42$^{de}$ | 1.15$^b$ | 1.32$^b$ | 1.80$^d$ |
| | | 40 | 17.75$^d$ | 55.05$^d$ | 112.50$^{cd}$ | 1.25$^b$ | 1.52$^b$ | 2.32$^{cd}$ |
| | | 60 | 15.67$^d$ | 47.95$^e$ | 109.50$^{cd}$ | 1.10$^c$ | 1.52$^b$ | 2.10$^{cd}$ |
| | | 80 | 11.55$^e$ | 49.72$^e$ | 102.07$^d$ | 0.90$^c$ | 1.50$^b$ | 1.97$^d$ |
| | Inoculated | 0 | 19.42$^{bc}$ | 74.95$^c$ | 115.75$^{cd}$ | 1.55$^a$ | 1.87$^b$ | 3.40$^{bc}$ |
| | | 20 | 20.85$^{bc}$ | 80.72$^c$ | 121.50$^c$ | 1.57$^a$ | 1.77$^b$ | 3.70$^{bc}$ |
| | | 40 | 34.15$^a$ | 97.37$^b$ | 143.00$^a$ | 1.77$^a$ | 2.35$^a$ | 4.50$^{ab}$ |
| | | 60 | 26.75$^{bc}$ | 84.65$^b$ | 127.32$^{ab}$ | 1.15$^b$ | 2.10$^{ab}$ | 4.10$^b$ |
| | | 80 | 13.25$^{de}$ | 42.07$^e$ | 112.82$^{cd}$ | 0.80$^c$ | 1.57$^b$ | 2.72$^{cd}$ |
| *M. pruriens* | Uninoculated | 0 | 10.60$^e$ | 63.10$^d$ | 103.00$^d$ | 0.35$^e$ | 1.10$^b$ | 2.22$^c$ |
| | | 20 | 16.25$^d$ | 74.25$^c$ | 112.25$^{cd}$ | 0.37$^e$ | 0.80$^b$ | 2.70$^{cd}$ |
| | | 40 | 22.47$^c$ | 88.45$^b$ | 117.00$^{cd}$ | 0.50$^{de}$ | 1.20$^b$ | 2.90$^{cd}$ |
| | | 60 | 22.07$^c$ | 87.87$^b$ | 109.50$^{cd}$ | 0.47$^{de}$ | 0.87$^b$ | 2.57$^{cd}$ |
| | | 80 | 21.80$^c$ | 83.50$^b$ | 103.75$^d$ | 0.47$^{de}$ | 0.67$^b$ | 1.95$^d$ |
| | Inoculation | 0 | 25.12$^{bc}$ | 91.65$^b$ | 121.75$^c$ | 0.80$^c$ | 1.50$^b$ | 3.45$^{bc}$ |
| | | 20 | 27.60$^{bc}$ | 71.05$^c$ | 133.00$^b$ | 0.65$^{de}$ | 1.77$^b$ | 3.80$^{bc}$ |
| | | 40 | 32.70$^a$ | 121.70$^a$ | 159.25$^a$ | 1.17$^b$ | 2.27$^a$ | 4.82$^{ab}$ |
| | | 60 | 27.97$^{bc}$ | 110.60$^a$ | 152.62$^a$ | 1.12$^{bc}$ | 2.05$^{ab}$ | 4.17$^b$ |
| | | 80 | 12.57$^{de}$ | 68.07$^d$ | 106.50$^{cd}$ | 0.52$^{de}$ | 1.12$^b$ | 1.82$^d$ |
| Significance level | | L×AMF | *** | *** | *** | *** | *** | *** |
| | | L×P | ** | *** | *** | ** | *** | *** |
| | | AMF×P | ** | *** | *** | *** | ** | *** |
| | | L×AMF×P | *** | *** | *** | *** | *** | *** |

**Notes.**

$^{a,b,c,d}$Means with different superscripts in the same column differ significantly at $P \leq 0.05$.

$^*P \leq 0.05$.

$^{**}P \leq 0.01$.

$^{***}P \leq 0.001$.

L, legume; AMF, Arbuscular mycorrhizal fungi; P, Phosphorous.

**Table 3  Interactions between legume species, AM fungi inoculation and P fertilizer level on chlorophyl content over time since sowing.**

| Independent variables | | | Chlorophyl content (° SPAD) | | | | |
|---|---|---|---|---|---|---|---|
| | | | Time elapsed since sowing (days) | | | | |
| Species | AMF | Fertilizer (kg P/ha) | 30 | 45 | 60 | 75 | 90 |
| *V. anguiculata* | Uninoculated | 0 | 45.1[d] | 53.1[de] | 63.2[de] | 67.8[cd] | 72.4[c] |
| | | 20 | 47.8[d] | 54.8[de] | 66.0[d] | 70.1[c] | 74.2[c] |
| | | 40 | 55.9[cd] | 58.7[d] | 69.1[d] | 73.2[c] | 77.9[b] |
| | | 60 | 48.5[cd] | 51.8[de] | 68.6[d] | 71.6[c] | 75.5[b] |
| | | 80 | 47.7[cd] | 50.4[e] | 63.4[d] | 67.5[cd] | 73.1[c] |
| | Inoculated | 0 | 57.2[cd] | 60.8[d] | 80.2[b] | 83.1[bc] | 86.4[ab] |
| | | 20 | 60.4[c] | 75.4[bc] | 82.7[ab] | 85.2[ab] | 88.3[ab] |
| | | 40 | 80.9[a] | 87.1[a] | 90.7[a] | 96.2[a] | 98.2[a] |
| | | 60 | 74.2[ab] | 80.2[ab] | 86.7[a] | 91.2[a] | 95.4[a] |
| | | 80 | 48.0[d] | 52.9[de] | 63.2[d] | 68.3[cd] | 74.4[c] |
| *L. purpereus* | Uninoculated | 0 | 40.6[d] | 45.1[f] | 50.4[f] | 54.2[e] | 58.6[d] |
| | | 20 | 41.8[d] | 47.7[f] | 53.4[ef] | 56.2[de] | 61.3[d] |
| | | 40 | 47.4[d] | 50.9[ef] | 56.9[e] | 60.2[de] | 63.5[d] |
| | | 60 | 44.6[d] | 49.1[f] | 54.9[e] | 57.2[de] | 62.4[d] |
| | | 80 | 42.0[d] | 43.8[f] | 49.2[f] | 53.2[e] | 60.3[d] |
| | Inoculated | 0 | 66.9[ab] | 70.7[c] | 75.9[b] | 80.2[bc] | 84.3[ab] |
| | | 20 | 70.0[ab] | 72.5[c] | 77.8[b] | 82.2[bc] | 87.2[ab] |
| | | 40 | 79.3[ab] | 82.7[ab] | 84.4[a] | 87.1[a] | 93.3[a] |
| | | 60 | 73.1[ab] | 76.6[bc] | 80.5[b] | 84.8[b] | 90.3[ab] |
| | | 80 | 48.5[cd] | 54.7[de] | 56.5[e] | 59.9[de] | 58.6[d] |
| *M. pruriens* | Uninoculated | 0 | 44.1[d] | 49.4[f] | 53.9[ef] | 58.2[de] | 62.4[d] |
| | | 20 | 46.7[cd] | 50.4[ef] | 55.9[e] | 60.2[de] | 64.3[d] |
| | | 40 | 55.0[c] | 58.5[de] | 60.0[e] | 64.3[de] | 68.4[d] |
| | | 60 | 46.6[cd] | 52.4[de] | 57.8[e] | 62.4[de] | 66.6[d] |
| | | 80 | 41.2[d] | 48.4[f] | 53.9[e] | 57.4[de] | 62.5[d] |
| | Inoculated | 0 | 49.4[c] | 56.2[d] | 61.4[de] | 66.4[de] | 70.4[c] |
| | | 20 | 50.0[c] | 58.5[d] | 64.3[c] | 68.3[cd] | 72.4[c] |
| | | 40 | 64.4[bc] | 74.0[bc] | 78.1[b] | 82.4[bc] | 86.4[ab] |
| | | 60 | 57.3[c] | 70.2[c] | 73.3[bc] | 79.8[bc] | 88.4[ab] |
| | | 80 | 42.2[d] | 50.4[e] | 49.8[f] | 58.8[de] | 64.5[d] |
| Significance level | | L×AMF | *** | *** | *** | *** | *** |
| | | L×P | ** | *** | *** | ** | *** |
| | | AMF×P | *** | ** | ** | *** | *** |
| | | L×AMF×P | *** | *** | *** | *** | *** |

**Notes.**
[a,b,c,d]Means with different superscripts in the same column differ significantly at $P \leq 0.05$.
*$P \leq 0.05$.
**$P \leq 0.01$.
***$P \leq 0.001$.
L, legume; AMF, Arbuscular mycorrhizal fungi; P, Phosphorous.

legume species, P fertilizer level and AMF inoculation were again significant ($p < 0.05$) for chlorophyll content. The AMF-inoculated plants showed interspecific responses to P fertilizer over time. AMF inoculated plants of *V. unguiculata* attained significantly ($p < 0.05$) higher chlorophyll content at 1.36 g P/pot (80.9–87.1 SPAD) and (74.2–80.2 SPAD) compared to other P fertilizer levels from day 30 to 45 post-sowing. For all P fertilizer levels, the AMF-inoculated *L. purpereus* plants had higher chlorophyll content than uninoculated plants until 90 days post-sowing. The chlorophyll content for the AMF inoculated *L. purpereus* was highest at 1.36 g P/pot (79.3–84.4 SPAD) and 2.05 g P/pot (73.1–80.5 SPAD), with plants grown at 2.72 g P/pot (48.5–56.5 SPAD) exhibiting lower chlorophyll content than plants grown in other P fertilizer levels until 90 days post-sowing. The results showed however, that the chlorophyll content of the AMF inoculated *L. purpereus* plants grown at 0.68 g P/pot was comparable ($p > 0.05$) to that of plants grown at 0.68 and 2.04 g P/pot on day 90 post-sowing. The remarkable responses of AMF-inoculated compared to uninoculated *M. pruriens* plants were evident 45 days post-sowing, more so at 1.36 g P/pot (74.0–88.4 SPAD) and 2.04 g P/pot (70.2–88.4 SPAD) compared to 0, 1.36 and 2.72 g P/pot.

## Leaf and stem yield

The interactions between P fertilizer, AMF-inoculation and legume species on leaf and stem yield and leaf to stem ratio are presented in Table 4. The three-way interaction between legume species, P fertilizer and AMF on leaf and stem dry matter production and leaf:stem ratio was significant ($p < 0.05$). For both AMF-inoculated and uninoculated *M. pruriens* plants, the leaf dry matter was significantly higher at 0.68 and 2.04 g P/pot compared to other P fertilizer levels. However, the former attained significantly ($p < 0.05$) higher leaf dry matter than the latter at 1.36 g P/pot (142.5 *vs* 87.4 g/pot) and 2.04 g P/pot (126.25 *vs* 84.17 g/pot) than other P fertilizer levels. Nonetheless, the leaf:stem ratio for AMF-inoculated *M. pruriens* was similar ($p > 0.05$) across the P fertilizer levels, with uninoculated plants grown at 0 and 2.72 g P/pot attaining remarkable higher leaf:stem ratio.

The AMF-inoculated *L. purpureus* plants produced significantly higher leaf and stem dry matter at certain P fertilizer levels. At 1.36 g P/pot, they yielded 123.25 g and 81.30 g/pot, respectively, and at 2.04 g P/pot, they yielded 106.25 g and 79.55 g/pot, respectively, compared to other P levels. However, at the highest P level (2.72 g P/pot), leaf dry matter in AMF-inoculated plants was similar to that of uninoculated plants. The leaf:stem ratio was highest in all P fertilizer levels for uninoculated plants relative to AMF-inoculated ones at 1.36 and 2.04 g P/pot. For *M. pruriens* also, the AMF-inoculated plants attained similar leaf dry matter of 142.5 and 126.25 g/pot at 1.36 and 2.72 g P/pot, respectively, which were significantly higher than that of uninoculated plants in all P fertilizer levels. However, the stem dry matter for AMF-inoculated plants was significantly higher at 1.36 g P/pot (120.00 g/pot) than all P fertilizer levels. However, the leaf:stem ratio was relatively low in AMF-inoculated *M. pruriens* grown at 1.36 and 2.72 g P/pot compared to uninoculated plants.

**Table 4  Interactions between legume species, AM fungi inoculation and P fertilizer level on dry matter production over time since sowing.**

| Species | AMF | Fertilizer (kg P/ha) | Leaf DM (g/pot) | Stem DM (g/pot) | Leaf to stem ratio |
|---|---|---|---|---|---|
| | | 0 | 83.32[e] | 43.47[gh] | 1.94[a] |
| | | 20 | 103.25[d] | 71.77[de] | 1.44[bc] |
| | Uninoculated | 40 | 121.75[bc] | 73.42[de] | 1.65[ab] |
| | | 60 | 114.50[bc] | 68.67[de] | 1.66[ab] |
| | | 80 | 82.40[ef] | 37.25[g] | 2.24[a] |
| *V. anguiculata* | | 0 | 104.00[cd] | 77.22[de] | 1.34[cd] |
| | | 20 | 122.75[bc] | 91.32[cd] | 1.34[cd] |
| | Inoculated | 40 | 144.00[a] | 114.62[a] | 1.26[cd] |
| | | 60 | 130.75[a] | 110.00[ab] | 1.19[d] |
| | | 80 | 102.50[d] | 95.17[bc] | 1.08[d] |
| | | 0 | 64.20[fg] | 30.32[h] | 2.12[a] |
| | | 20 | 70.47[f] | 37.40[gh] | 1.88[ab] |
| | Uninoculated | 40 | 75.22[f] | 35.7[gh] | 2.10[a] |
| | | 60 | 73.60[f] | 36.10[gh] | 2.04[a] |
| | | 80 | 69.15[f] | 33.20[gh] | 2.09[a] |
| *L. purpereus* | | 0 | 86.30[de] | 44.25[gh] | 1.95[ab] |
| | | 20 | 87.60[de] | 47.27[fg] | 1.86[ab] |
| | Inoculated | 40 | 123.25[bc] | 81.30[cde] | 1.52[bc] |
| | | 60 | 106.25[cd] | 79.55[de] | 1.34[cd] |
| | | 80 | 71.60[ef] | 48.77[fg] | 1.46[bc] |
| | | 0 | 75.62[ef] | 38.00[gh] | 1.99[a] |
| | | 20 | 79.62[ef] | 44.62[g] | 1.78[ab] |
| | Uninoculated | 40 | 87.40[de] | 51.45[f] | 1.71[ab] |
| | | 60 | 84.17[de] | 53.80[f] | 1.57[ab] |
| | | 80 | 78.75[ef] | 45.45[fg] | 1.73[ab] |
| *M. pruriens* | | 0 | 95.12[de] | 53.60[f] | 1.77[ab] |
| | | 20 | 98.82[de] | 59.15[ef] | 1.67[ab] |
| | Inoculated | 40 | 142.50[a] | 120.00[a] | 1.18[d] |
| | | 60 | 126.25[ab] | 92.32[c] | 1.36[cd] |
| | | 80 | 87.75[de] | 46.35[f] | 1.89[ab] |
| Significance level | | L×AMF | ** | ** | * |
| | | L×P | ** | ** | ** |
| | | AMF×P | *** | *** | ** |
| | | L×AMF×P | *** | *** | ** |

Notes.

[a,b,c,d]Means with different superscripts in the same column differ significantly at $P \leq 0.05$.

*$P \leq 0.05$.

**$P \leq 0.01$.

***$P \leq 0.001$.

L, legume; AMF, Arbuscular mycorrhizal fungi; P, Phosphorous.

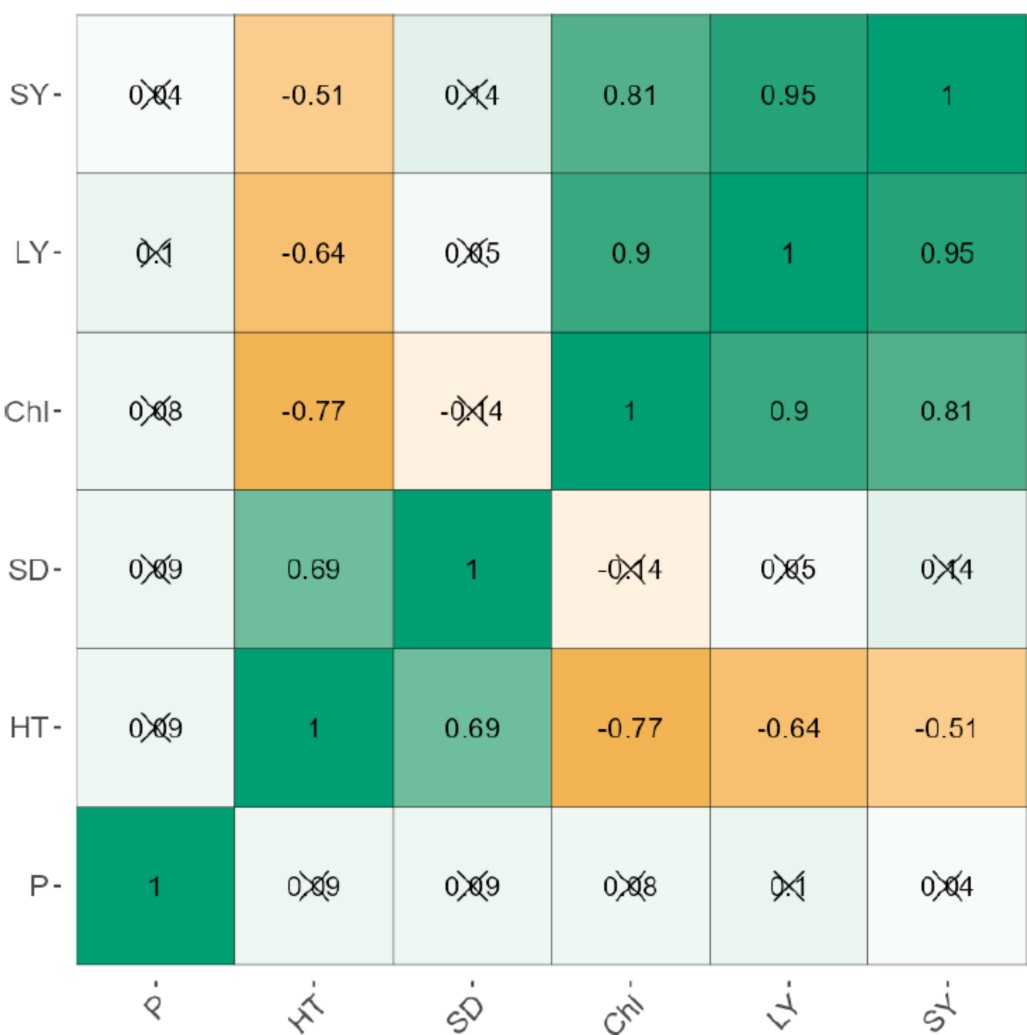

**Figure 1** The correlation between the agronomic traits of uninoculated forage legumes grown under varying levels of Phosphorous.

## The correlations between the agronomic traits of forage legume species

The leaf and stem yield were negatively correlated significantly ($p < 0.001$) for uninoculated ($r = 0.95$) and inoculated plants ($r = 0.93$; Figs. 1 and 2). Chlorophyll content also increased significantly ($P < 0.001$) with stem and leaf yield ($r = 0.64$–$0.76$) for uninoculated plants (Fig. 1). Similarly, chlorophyll content was positively related to leaf and stem yield of inoculated plants (Fig. 2). Plant height exhibited weak negative correlations with leaf yield ($r = -0.51$), stem yield ($r = -0.64$), and chlorophyll content ($r = -0.77$). Phosphorus levels showed no significant correlation with any agronomic trait in both inoculated and uninoculated plants (Figs. 1 and 2).

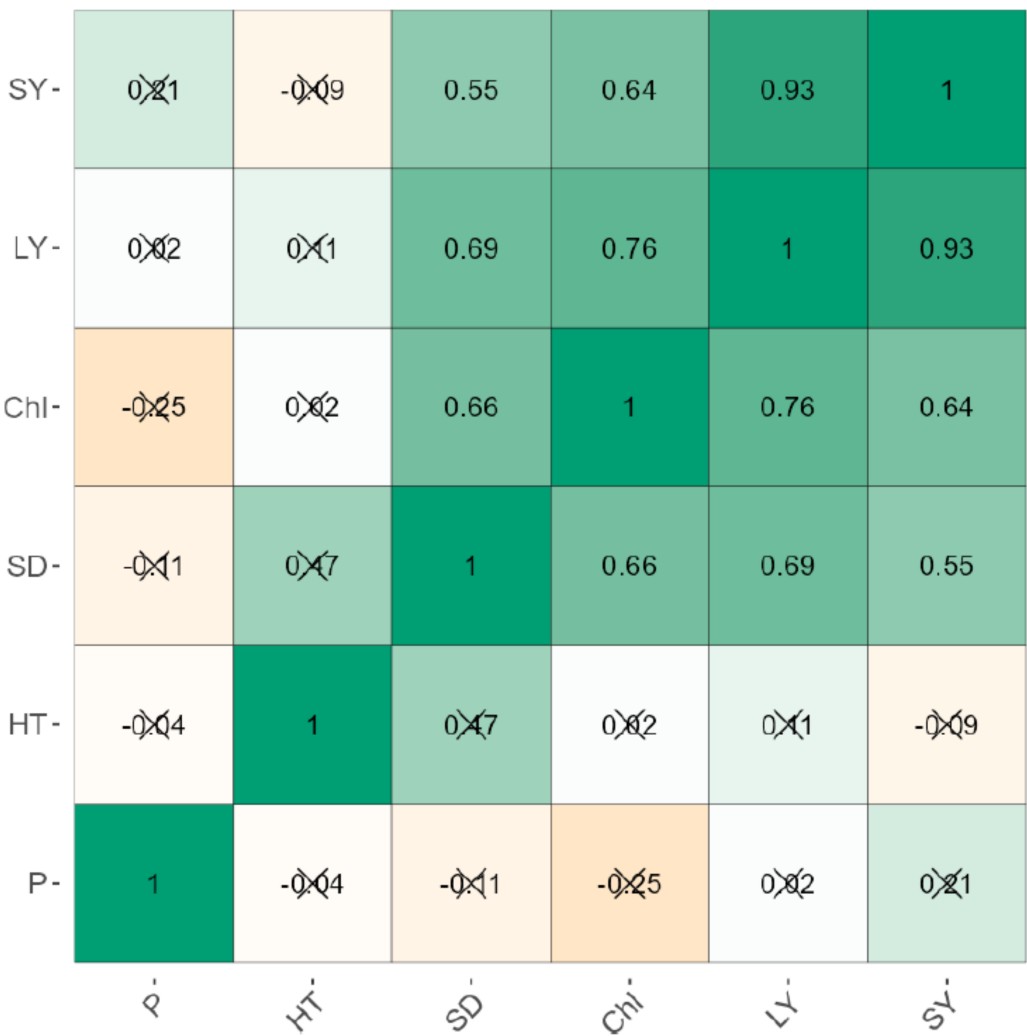

**Figure 2** The correlation between the agronomic traits of inoculated forage legumes grown under varying levels of Phosphorous.

## DISCUSSION

### Shoot height and stem diameter

The interactions between legume species, arbuscular mycorrhizal fungi (AMF) and P fertilizer level suggest that legume growth and chlorophyll content responses are determined by synergistic effects of P fertilizer and AMF depending on the type of legume species. This highlights that the crucial role of AMF for efficient utilization of P by legumes also depends on the amount of P in the soil. This was justified by high legume growth and chlorophyll content at 1.36 g P/pot, above which the stimulatory effect of AMF was negated in all legume species, regardless of the time elapsed since sowing (Table 2). Indeed, at 2.72 g P/pot, inoculated legumes exhibited a stunted growth (Table 2), indicating that growth stimulation by AMF is limited in soils with excess P. Similarly, *Liu et al. (2020)* and *Xia et al. (2023)* found similar responses in which chlorophyll content and productivity

of Alfalfa increased with an increased P rate to some degree and declined at relatively high P level. As indicated previously by *Nwaga, Ambassa-kiki & Nsangou (2003)*, *Khan et al. (2008)*, *Nishita & Joshi (2010)*, and *Tobisa & Uchida (2017)*, the relatively high soil P reduces the symbiotic association between legumes and AMF, with AMF tending to be parasitic to the host plant at higher P levels. Even at 0–0.68 g P/pot, growth enhancement by AMF-inoculation was minimal, probably due to P deficiency. This finding disagrees with several previous studies *e.g.*, *Yaseen, Burni & Hussain (2011)*, *Nazir et al. (2011)*, *Singh & Yadav (2008)*, *Nouri et al. (2014)*, and *Chen et al. (2023)*, that suggest that AMF compensates for low soil P *via* enhancing the acquisition of other nutrients. It can therefore be, deduced that the enhancement of productivity and growth of legumes by AMF depends on the optimal rate of P fertilizer supply, which in this study appears to be 1.36 g P/pot. In congruence with our findings, *Dhillon & Vig (1996)* also found that 1.36 g P/pot in a $P_2O_5$ form optimized forage yield of a leguminous species (*Vigna radiata*). Although not examined in this study, AMF colonization rate and diversity are constrained at excessive P application, thereby negating the responses of biomass and growth of legumes (*Xia et al., 2023*).

## Chlorophyll content

The higher P level (2.72 g P/pot) dramatically reduced growth, productivity and chlorophyll content of almost all legumes studied here. The systematic review of *Mitran et al. (2018)*, however, indicates that the optimal rate of P fertilizer is species- and area-specific, depending largely on the P status of the soil. Thus, in soils highly deficient in P, the optimal rate of P application may be higher than in soils rich in P (*Mitran et al., 2018*). The significant three-way interactions also highlighted that growth responses of legumes do not only vary by AMF-inoculation and P fertilizer level, but also interspecifically. Of the tested inoculated legume species, *Mucuna pruriens* was generally more responsive to 1.36 g P/pot relative to *Vigna unguiculata* and *Lablab purpereus* (Table 3), suggesting that the former utilizes P more efficiently than the latter two species. These findings form the basis for species selection for legume pasture establishment and production. Generally, AMF-inoculation enhances P use efficiency in plants, thereby enhancing growth (*Ibrahim et al., 2022*). Because P is not readily available to plants, AMF *via* producing enzymes that solubilize P, induces higher uptake of P in inoculated plants (*Chen et al., 2023*). Apart from enhancing P uptake, AMF enhances the uptake of other essential nutrients (*e.g.*, N, K and Fe) and reduces the uptake of salt ions, thereby stimulating vigorous plant growth and photosynthesis (*Begum et al., 2019*). We found, however, that for inoculated *Vigna unguiculata* and *Lablab pupereus* plants, the differences in chlorophyll content from 0–2.04 g P/pot disappeared with time since sowing (Table 3). This could imply that despite low P content at 0 and 0.68 g P/pot, as plants grew, they were able to acquire more P and efficiently channel it to photosynthetic apparatus. This was further justified by a positive correlation between chlorophyll content, leaf and stem yield (Figs. 1 and 2). Generally, an increase in chlorophyll content increases leaf area and size, thereby increasing forage yield. However, the correlation results indicated that chlorophyll content and leaf yield were negatively related to plant height (Figs. 1 and 2), indicating that P was channeled to

and efficiently used for photosynthesis and leaf production rather than stem elongation. Similarly, *Guler & Ozcelik (2007)* found a positive relationship between chlorophyll content and productivity of *Phaseolus vulgaris*. The results of this study are in part congruent with *Zaeem et al. (2019)*, who found a strong correlation between chlorophyll content and productivity, but differ in that chlorophyll content declined with increase in plant height. Our results suggest that chlorophyll content is a major determinant of forage production, regardless of plant height.

As plants matured at high P levels (1.36–2.04 g P/pot), the pot size possibly limited further root growth and biomass production, thereby downregulating photosynthesis. Nonetheless, higher leaf to stem ratio for inoculated plants at low P fertilizer levels of 0 and 0.68 g P/pot (Table 3), suggested that the little available P was invested in leafiness rather than stem production. Generally, as legume plants grow, especially AMF-inoculated plants with their tap root system, have better access to nutrients in the deeper zones of the soil profile. For all legume species, however, the chlorophyll content remained different only at 2.72 g P/pot compared to other P fertilizer levels, implying that excessive P does not only stunt legume growth, but also inhibits photosynthesis.

## Leaf and stem yield

Both the leaf and stem biomass were highest in inoculated than uninoculated plants, with these responses being remarkable at 1.36 g P/pot (Table 4). This is not surprising, given that the plant growth was higher in AMF-inoculated compared to uninoculated plants at these P fertilizer levels. This could be ascribed to the fact that P was optimal to permit high stimulation of legume productivity by AMF *via* increased photosynthesis and growth at 1.36 g P/pot relative to uninoculated plants. Enhancement of legume growth by AMF-inoculation has been reported in other studies investigating legume responses to interactive effects of AMF and P fertilizer. However, the leaf:stem ratio was highest for uninoculated relative inoculated plants (Table 4), suggesting that the latter invested more P not only in leaf production, but also in stem production. This has significant implications for animal nutrition, as the higher stem production translates to low forage quality due to low crude protein and high fiber content in stems relative to leaves (*Mganga et al., 2021*). The forage material with higher stem component has a longer retention time in the rumen due to low digestibility (*Mganga et al., 2021*). It should be noted, however, that despite low leaf:stem ratios for inoculated plants, they still exhibited more leafiness than uninoculated plants. Thus, a well-timed grazing management will be needed to ensure that these legumes are utilized while they are still nutritious before their stems become more fibrous and ligneous.

## Study limitations and future research prospects

This study was conducted in the glasshouse using pot experiment. While the AMF inoculation enhances root growth and high development of hyphae uptake of P and other essential nutrients (*Chen et al., 2023*), root growth in pots is likely to be restricted by pot size (*Chalk et al., 2006*). This has serious implications on the above-ground responses of legumes, as the inhibition of tap root growth may limit shoot growth and productivity.

As a result, it is not surprising that chlorophyll content did not differ from day 75 to day 90 post-sowing (Table 3). This could be due to downregulation of photosynthesis because of restricted vertical growth of tap roots and low root biomass production due to limited carrying capacity of pots (Qin et al., 2022; Mndela et al., 2022). Apart from the pot size, glasshouse conditions do not mimic wide range of climate scenarios, thus we caution that the applicability of these results is restricted to areas in which climate conditions resemble those set at the glasshouse in this study. Due to financial constraints, the root biomass and AMF colonization rate were not determined along the P fertilizer gradient studied here, instead these parameters were investigated only at optimal P level (1.36 g P/pot) under field conditions (Mpongwana et al., 2023b). This, therefore, limits the understanding of how the interaction of AMF and P fertilizer influences below-ground processes along a range of P fertilizer levels. Thus, to bridge this gap, future research should be directed towards understanding how AMF interacts with P to influence below-ground responses and how these responses feedback to above-ground shoot responses under field conditions. This research may include an assessment of the effect of AMF on root growth, rooting depth and productivity and mycelial biomass. This research may further investigate how belowground productivity including root and mycelial productivity influences P use efficiency and nutrient uptake of AMF-inoculated *vs* uninoculated plants.

## CONCLUSION

This study provides evidence that AMF-inoculation is key in enhancing legume growth, and productivity. Our results show, however, that the influence of AMF depends largely on its interaction with P fertilizer and legume species. For instance, remarkable responses of forage legumes in terms of growth, chlorophyll content and dry matter production to AMF-inoculation were noticeable at 1.36 g P/pot, implying that this P fertilizer level is optimal for legume growth and productivity. However, it was noted that responses to AMF-inoculation and P fertilizer were interspecific, emphasizing the importance of appropriate species selection for pasture establishment. These findings are a basis for soil nutrition management of forage legume pastures and may play a crucial role in policy making concerning pasture establishment in communal arable lands of South Africa. Therefore, the findings suggest that the incorporation of AMF-inoculation in pasture establishment can ensure high plant growth and productivity, particularly under the optimal rate of P fertilizer application (1.36 g P/pot). Our results also showed that *Mucuna pruriens* was the highly responsive legume in terms of growth and productivity, highlighting the significance of this species in pasture establishment.

### Funding

This work was funded by the National Research Foundation (T353) of South Africa and the Govan Mbeki Research and Development Centre (C203) (South Africa). The funders had

no role in study design, data collection and analysis, decision to publish, or preparation of the manuscript.

## Grant Disclosures

The following grant information was disclosed by the authors:
National Research Foundation (T353) of South Africa.
Govan Mbeki Research and Development Centre: C203.

## Competing Interests

The authors declare there are no competing interests.

## Author Contributions

- Sanele Mpongwana conceived and designed the experiments, performed the experiments, analyzed the data, prepared figures and/or tables, authored or reviewed drafts of the article, and approved the final draft.
- Alen Manyevere conceived and designed the experiments, performed the experiments, analyzed the data, prepared figures and/or tables, authored or reviewed drafts of the article, and approved the final draft.
- Conference Thando Mpendulo conceived and designed the experiments, performed the experiments, analyzed the data, prepared figures and/or tables, authored or reviewed drafts of the article, and approved the final draft.
- Johnfisher Mupangwa conceived and designed the experiments, performed the experiments, analyzed the data, authored or reviewed drafts of the article, and approved the final draft.
- Wandile Mashece performed the experiments, analyzed the data, authored or reviewed drafts of the article, and approved the final draft.
- Mthunzi Mndela performed the experiments, analyzed the data, prepared figures and/or tables, authored or reviewed drafts of the article, and approved the final draft.

## Data Availability

   The data is available in the Supplementary File.

## Supplemental Information

Supplemental information for this article can be found online at http://dx.doi.org/10.7717/peerj.18955#supplemental-information.

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
